# Clinical Relevance of Red Blood Cell Distribution Width (RDW) in Endometrial Cancer: A Retrospective Single-Center Experience from Korea

**DOI:** 10.3390/cancers15153984

**Published:** 2023-08-05

**Authors:** Kyung-Jin Eoh, Tae-Kyung Lee, Eun-Ji Nam, Sang-Wun Kim, Young-Tae Kim

**Affiliations:** 1Department of Obstetrics and Gynecology, Yongin Severance Hospital, College of Medicine, Yonsei University, Yongin 16995, Republic of Korea; kjeoh2030@yuhs.ac; 2Department of Obstetrics and Gynecology, College of Medicine, Inha University, Incheon 22332, Republic of Korea; gooddoc0901@gmail.com; 3Department of Obstetrics and Gynecology, Institute of Women’s Medical Life Science, Severance Hospital, College of Medicine, Yonsei University, Seoul 06273, Republic of Korea; nahmej6@yuhs.ac (E.-J.N.); san1@yuhs.ac (S.-W.K.)

**Keywords:** endometrial neoplasms, red blood cell distribution, mortality

## Abstract

**Simple Summary:**

This study aims to investigate the potential use of the red blood cell distribution width (RDW) as a biomarker for predicting prognosis and recurrence in patients with endometrial cancer. We believe that our study makes a significant contribution to the literature because it presents a retrospective analysis of clinical data from 431 patients diagnosed with endometrial cancer, which was conducted to investigate the association between RDW and survival outcomes. The study found that patients with high RDW values had significantly worse overall survival (OS) and disease-free survival (DFS) compared to those with low RDW values. RDW was also found to be an independent prognostic factor for OS and DFS. Further, we believe that this paper will be of interest to the readership of this journal because the study identifies a simple and cost-effective biomarker that can aid in the prognostic evaluation of patients with endometrial cancer.

**Abstract:**

Background: Red blood cell distribution width (RDW) is a standard parameter of complete blood count and indicates the variability in red blood cell size. This study aimed to determine whether preoperative RDW can be used to predict the recurrence and prognosis of endometrial carcinoma. Methods: The medical records of 431 patients diagnosed with endometrial carcinoma were retrospectively reviewed between May 2006 and June 2018. In addition to RDW, the clinicopathological factors, survival curves, and prognoses of the patients with endometrial carcinoma were compared between the high (*n* = 213) and low (*n* = 218) groups according to the median RDW value (12.8%). Results: The patients with high RDW had significantly advanced-stage (*p* = 0.00) pelvic lymph node metastasis (*p* = 0.01) and recurrence (*p* = 0.01) compared to those in the low-RDW group. In univariate analysis with DFS as the endpoint, surgical stage, type II histology, grade, RDW, and lymph node metastasis were independently associated with survival. Patients with high RDW values had significantly shorter disease-free survival and overall survival than those with low RDW values (log-rank *p* = 0.03, log-rank *p* = 0.04, respectively). Conclusion: Our results demonstrate that RDW is a simple and convenient indicator of endometrial carcinoma recurrence. Prospective studies are needed to validate the findings of the current study.

## 1. Introduction

Endometrial cancer is the second most prevalent gynecological malignancy worldwide, with an overall favorable prognosis. However, some patients have a high risk of recurrence and a poor prognosis. Surgical staging is a pivotal step in the management of endometrial cancer. The pathological evaluation of surgical specimens can determine the need for postoperative adjuvant treatment to minimize recurrence [1,2]. The clinical performance of patients, including their hematological parameters, is fundamentally linked to surgery.

The scientific literature has demonstrated that cancer progression and survival outcomes are influenced by various factors, in addition to tumor characteristics. Patient age, clinical performance, nutrition, and hematological status play significant roles in determining their outcomes [3,4]. Systemic inflammation associated with cancer is a well-known factor that affects patient prognosis for different types of cancer [5,6,7].

Red cell distribution width (RDW) is a hematological parameter that measures the variation in the size of circulating red blood cells (RBCs), reflecting alterations in RBC survival patterns and indicating disturbances in erythropoiesis. Recent studies have shown that elevated RDW is associated with persistent inflammation and that it serves as a prognostic marker in various cancers [8,9,10]. Notably, there is increasing evidence linking cancer-related inflammation with increased RDW levels, which is thought to be driven by the release of cytokines such as interleukin-6 and tumor necrosis factor-alpha [11,12]. These findings suggest that monitoring RDW changes may provide valuable insights into cancer progression and its associated inflammatory responses.

It is reasonable to hypothesize that RDW may aid in predicting the prognosis of patients with cancer. Despite the potential clinical utility of RDW, its application in oncology remains relatively unexplored. The scarcity of publications evaluating the clinical significance of RDW alone on the survival outcome of endometrial cancer in the literature prompted the need for this single-center study. Furthermore, only a limited number of studies have specifically examined the relationships among RDW, cancer progression, and patient survival in the context of endometrial cancer. The objective of this study was to determine the clinical relevance of preoperative red cell distribution width (RDW) in predicting the risk of recurrence and overall prognosis of endometrial carcinoma.

## 2. Materials and Methods

### 2.1. Patients

Clinical and demographic information was obtained from the medical data of the electronic Severance Hospital platform. We retrospectively reviewed patients with Federation of Gynecology and Obstetrics (FIGO) stage I–IV endometrial cancer who underwent open staging surgery or robot-assisted staging at the Yonsei Cancer Center, Seoul, Republic of Korea, between May 2006 and June 2018. Women aged 18 years and older were included in the study. From the electronic medical records, we identified clinical variables such as age, BMI, histology, grade, FIGO stage, harvested and metastatic lymph nodes, red blood cell count, red blood cell distribution width levels, hemoglobin levels, intra-operative blood loss, post-operative transfusion, and the use of adjuvant therapy. All patients underwent procedures such as an extrafascial hysterectomy with bilateral salpingo-oophorectomy, bilateral pelvic lymph node dissection, para-aortic lymph node dissection, or peritoneal cytology examination. A Foley catheter was inserted to empty the bladder. The surgical assistants at the bedside and caudal parts of the patient were usually chief residents or training fellows. Robot-assisted staging surgery was performed using a da Vinci Telerobotic System (Intuitive Surgical, Inc. Sunnyvale, CA, USA). All three systems (S, Si, and Xi [Intuitive Surgical, Inc.]) were used during the study period.

Blood samples for routine blood tests were collected via standard venipuncture of the peripheral veins of patients with endometrial cancer 24–72 h before surgical staging. White blood cell levels, red blood cell count, red blood cell distribution width levels, and hemoglobin levels were measured with the Sysmex XN (Tokyo, Japan) automatic blood counting system within 0.5 h after blood collection. RDW value was defined as the coefficient of variation (percentage) in red blood cell volume, which measures the degree of variability in the size of red blood cells. In this study, we considered the RDW level to be high when it was over 12.8% (the median value). Blood samples were collected for biochemical analyses.

### 2.2. Statistical Analysis

Our institutional follow-up strategy was to monitor patients every 3 months for the first 2 years after treatment and every 6 months thereafter. Recurrence was defined as the date of appearance of a disease detected radiologically or histologically during a follow-up examination.

Disease-free survival (DFS) was defined as the time interval between the date of initial diagnosis and disease progression based on the Response Evaluation Criteria in Solid Tumors (version 152 1.1). We calculated the overall survival (OS) as the time interval between the date of initial diagnosis and cancer-related death or the end of the study.

All statistical analyses were performed using IBM SPSS version 25 for Windows (IBM Corp., Armonk, NY, USA). According to preoperative RDW values, patients were divided into two groups: a high-RDW group (>12.8%) and a low-RDW group (≤12.8%). Descriptive statistics were used for the demographic data and are summarized as mean (standard deviation) or frequency (percentage). The Kolmogorov–Smirnov test was used to verify standard normal distributional assumptions. Student’s t-test and the Mann–Whitney U test were used for parametric and non-parametric variables, respectively. The differences in patient characteristics between the groups were compared using chi-square or Fisher’s exact tests with respect to time intervals. DFS and OS were analyzed using Kaplan–Meier curve analysis, and the groups were compared using the log-rank test. The Cox proportional hazards regression model was employed for both univariate and multivariate analyses. Statistical significance was set at *p* < 0.05.

## 3. Results

A total of 431 patients with endometrial cancer were included in this study. The patient characteristics of the study population are shown in Table 1.

The preoperative median of RDW in this study was 12.8%, with a range from 11.1% to 27.8%. Patients were divided into two groups based on the median RDW value, with 213 patients having high RDW (>12.8%) and 218 patients having low RDW (≤12.8%). Significant differences were found in age (*p* = 0.01), BMI (*p* = 0.01), FIGO stage (*p* =0.03), number of pelvic metastatic lymph nodes (*p* = 0.01), and recurrence (*p* = 0.01) between the two groups. Kruskal–Wallis analysis revealed a significant difference for the proportion of patients with high RDW, as this was higher in stages II, III, and IV than in stage I. The percentages of patients with high RDW values were 14.4%, 32.2%, 37.6%, and 39.4% for stages I (*n* = 331), stage II (*n* = 22), stage III (*n*-56), and stage IV patients (*n* = 22), respectively (Figure 1).

The median follow-up period after staging surgery was calculated to be 49.5 months (range: 12–124 months). The five-year DFS rates were 87% and 92% in the high-RDW group and low-RDW group, respectively. The five-year OS rates were 89% and 96% in the high RDW-group and low-RDW group, respectively. Patients with high RDW values had significantly shorter disease-free survival and overall survival than those with low RDW values (log-rank *p* = 0.03, log-rank *p* = 0.04, respectively) (Figure 2).

In the univariate regression analyses of the complete cohort with DFS as the endpoint, FIGO stage, RDW, grade, histology, and lymph node metastasis were associated with an increased risk of recurrence, whereas in the multivariate analysis, RDW (*p* = 0.03) and grade 3 (*p* < 0.00) were significant independent risk factors. With OS as the endpoint, FIGO stage, RDW, histology, grade, and lymph node metastasis were found to be significantly associated with a worse prognosis in the univariate regression analysis. In the multivariate analysis, FIGO stage IV (*p* < 0.00), RDW (*p* = 0.04), and grade 3 (*p* = 0.02) were identified as significant independent risk factors. (Table 2).

## 4. Discussion

The current study demonstrated that elevated preoperative RDW values were significantly correlated with poor prognosis in patients with endometrial cancer. To the best of our knowledge, this is the first study to evaluate the clinical relevance of RDW alone in the prognosis of patients with endometrial cancer. The present study indicated that patients with higher RDW values had worse survival outcomes than those with lower RDW values. Kaplan–Meier cumulative survival analysis for DFS and OS demonstrated that high RDW values indicated significantly shorter DFS and OS in patients with endometrial cancer. These results revealed that RDW may play a pivotal role in predicting the survival of patients with endometrial cancer, particularly those with advanced disease. This is consistent with previous findings in patients with other malignancies, including breast, lung, and gynecologic cancers [13,14,15,16,17].

Takeuchi et al. performed a retrospective study on breast cancer in 299 patients and found a correlation between a higher RDW-to-platelet ratio and lower OS [16]. Li et al. demonstrated a significant correlation between higher RDW and neutrophil-to-lymphocyte ratio and worse overall survival [14]. Regarding endometrial cancer, Zhong et al. reported that RDW plus CA-125 was a significant independent prognostic factors for overall survival [17]. These researchers evaluated prognosis by combining RDW and various parameters, including the platelet ratio, neutrophil-to-lymphocyte ratio, and CA-125. They did not report the clinical relevance of evaluating prognosis based on RDW alone for an accurate prognostic assessment. Furthermore, this results of this study revealed that preoperative RDW values were significantly correlated with the FIGO stage, BMI, lymph node involvement, and recurrence in patients with endometrial cancer. The high-RDW group had a more advanced tumor stage, more extensive lymph node disease, and more frequent recurrence than cases with low RDW values. These findings support the hypothesis that elevated RDW levels may indicate tumor-related systemic inflammation due to more aggressive staging surgery in patients with obesity. Notably, our results also suggest that preoperative RDW values are associated with a high BMI, particularly in patients with advanced-stage disease. Obesity has been extensively linked to poor inflammation and worse clinical comorbidities, including diabetes, hypertension, and cancer. The condition leads to the creation of a proinflammatory milieu that is characterized by high levels of C-reactive protein, interleukin-6 (IL-6), and tumor necrosis factor-α in circulation. Additionally, obesity results in a relative deficiency of protective immune cell types, which may contribute to impaired immune dysregulation [18]. Consistent with this hypothesis, we also observed a significant association between high RDW and unfavorable disease-free survival and overall survival. This finding is consistent with the results of other studies on different cancers [8,19,20,21]. 

RDW is an indicator of the differential diagnosis of anemia. It is calculated by dividing the standard deviation of red blood cell size by the mean corpuscular volume and shows the variability in the size of circulating RBCs [22,23]. The calculation of RDW is an inexpensive test, and its value is routinely reported by automated analyzers used to perform the complete blood count (CBC). In addition to the differential diagnosis of anemia, high RDW levels have also been correlated with cardiovascular and renal diseases [24,25]. Studies have suggested that higher RDW values are significantly correlated with increased mortality in the general population and patients with septicemia and hepatitis [26,27]. The underlying mechanism has not been evaluated; however, high RDW values reflect chronic inflammation and increased levels of circulating cytokines, including IL-6, and hepsidin [12,28]. Anisocytosis, a condition characterized by varying sizes of red blood cells, has historically been neglected in prognostic assessments. However, recent studies have unveiled the promising potential of RDW as a biomarker of poor prognosis in numerous cancer types [20,29]. This may be attributed to the association between high RDW values and chronic inflammation as well as disruptions in erythropoiesis. During tumorigenesis, the local advancement of the tumor typically triggers an inflammatory response and subclinical bleeding. Hence, RDW can serve as an efficient marker of systemic inflammation and is now considered the seventh hallmark of cancer [30]. As subclinical bleeding and chronic inflammation progress and intensify, RDW may be a straightforward and cost-effective addition to traditional evaluation tools, enabling reliable prognostic assessment of patients with endometrial cancer.

The correlation between hematological parameters and cancer prognosis is a complex and multifaceted topic. There are some reports that suggest that other hematological parameters can be correlated with the prognosis of endometrial cancer. The prognostic significance of other parameters, including preoperative leukocytosis, anemia, and thrombosis, has been reported through the use of high-quality research designs such as systematic review and meta-analysis reports in endometrial cancer [31,32].

A strength of the current study was that all surgical staging and adjuvant treatments were performed at a tertiary referral institution by gynecologic oncologists and designated radiation oncologists. Moreover, the findings of this study hold practical implications for practitioners, as they suggest the potential utility of RDW as a prognostic biomarker for endometrial cancer. Despite these strengths, our study had some limitations, such as the retrospective nature of the clinical study and unmeasured variables that can cause confounding factors. Moreover, potential selection bias, especially due to the selection of patients who could undergo several types of adjuvant therapies, may also exist. We also acknowledge the fact that our study was a single-center study, which could affect the generalizability of our findings. Additionally, the single-group design restricts our capacity to establish definitive cause-and-effect relationships, while the lack of five-year follow-up data for all patients further adds to the list of limitations of this research study. We need to compare the sensitivity/specificity of RDW as a prognostic marker in a prospective setting.

## 5. Conclusions

Our findings suggest that RDW may serve as a useful prognostic biomarker for patients with endometrial cancer, enabling clinicians to identify high-risk individuals and initiate appropriate interventions. Additional studies are needed to validate the findings of the current study.

## Figures and Tables

**Figure 1 cancers-15-03984-f001:**
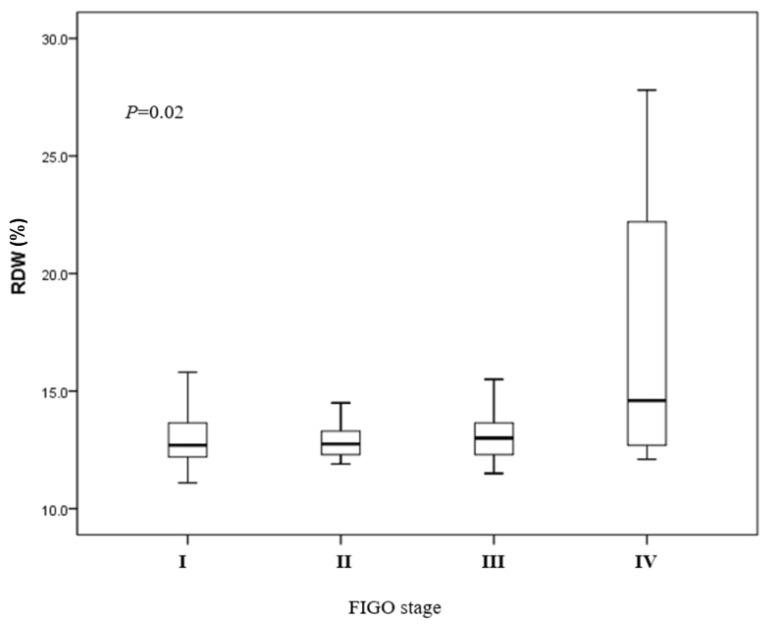
RDW values and FIGO stage. In each box, the lower and upper ends show the 25th and 75th percentiles, respectively. Capped bars demonstrate the minimum and maximum values, respectively. The line inside the box reveals the median RDW value.

**Figure 2 cancers-15-03984-f002:**
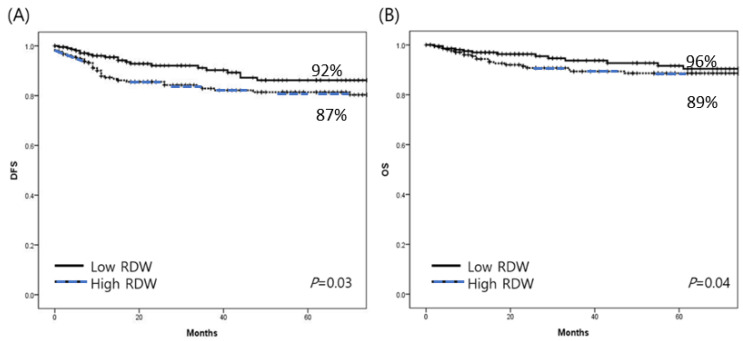
(**A**) Disease-free survival and (**B**) overall survival in patients with endometrial cancer. Five-year DFS were 92% and 87% in the low- and high-RDW groups, respectively. Five-year OS percentages were 96% and 89% in the low- and high-RDW groups, respectively.

**Table 1 cancers-15-03984-t001:** Patient characteristics of the study population.

Characteristics	Total Patients	RDW		
		Low (*n* = 218)	High (*n* = 213)	*p*
Age, mean (SD)	53.8 (9.3)	55.9 (8.1)	51.7 (10.5)	0.01
BMI, kg/m² (SD)	24.8 (5.1)	24.6 (3.8)	25.0 (6.3)	0.00
Preoperative Hb (SD)	12.1 (1.6)	12.6 (1.5)	11.6 (1.8)	0.11
FIGO stage *n*, (%)				0.03
I	331	177 (81.2)	154 (72.3)	
II	22	13 (6.0)	9 (4.2)	
III	56	21 (9.6)	35 (16.4)	
IV	22	7 (3.2)	15 (7.0)	
Cell type, *n* (%)				0.61
Endometrioid	317	158 (72.5)	159 (74.6)	
Other	114	60 (27.5)	54 (25.4)	
Grade, *n* (%)				0.34
1	186	91 (41.7)	95 (44.6)	
2	134	75 (34.4)	59 (27.7)	
3	111	52 (23.9)	59 (27.7)	
Metastatic LN, *n* (%)				
Pelvic LN	55	19 (8.7)	36 (16.9)	0.01
Paraaortic LN	32	16 (7.3)	16 (6.1)	0.61
EBL, mL (SD)		162.6 (217.9)	320.4 (612.9)	0.00
Adjuvant therapy				
Radiation	84	49 (22.5)	35 (16.4)	0.18
Chemotherapy	89	37 (17.0)	52 (24.4)	
Both	29	15 (6.9)	14 (6.6)	
None	229	117 (53.7)	112 (52.6)	
Recurrence, *n* (%)				0.01
No	374	198 (90.8)	176 (82.6)	
Yes	57	20 (9.2)	37 (17.4)	
Vital status, *n* (%)				0.05
Alive	394	205 (94.0)	189 (88.7)	
Dead	37	13 (6.0)	24 (11.3)	

RDW, red blood cell distribution width; SD, standard deviation; BMI, body mass index; LN, lymph node; EBL, estimated blood loss.

**Table 2 cancers-15-03984-t002:** Univariate and multivariate analysis of the complete cohort (*n* = 431) with disease-free survival and overall survival as the end points.

	DFS				OS			
Variables	Univariate Analysis		Multivariate Analysis		Univariate Analysis		Multivariate Analysis	
	HR (95% CI)	*p*	HR (95% CI)	*p*	HR (95% CI)	*p*	HR (95% CI)	*p*
Age								
≤49	1.00				1.00			
≥50	1.24 (0.68–2.27)	0.48			2.62 (1.02–6.72)	0.06		
FIGO stage								
I	1.00		1.00		1.00		1.00	
II	3.84 (1.57–9.36)	0.03	3.19 (1.26–8.05)	0.01	2.96 (0.86–10.16)	0.09	2.17 (0.62–7.61)	0.23
III	3.56 (1.82–6.98)	0.00	1.40 (0.50–3.95)	0.53	3.09 (1.26–7.55)	0.01	1.05 (0.25–4.43)	0.95
IV	18.79 (9.52–37.09)	0.00	7.73 (2.78–21.50)	0.00	20.01 (9.10–43.99)	0.00	7.99 (2.07–30.90)	0.00
RDW								
Low	1.00		1.00		1.00			
High	1.80 (0.84–3.10)	0.03	1.51 (0.84–2.72)	0.04	1.73 (0.88–3.41)	0.04	1.63 (0.78–2.41)	0.04
Histology								
Endometrioid	1.00		1.00		1.00		1.00	
Other	3.02 (1.78–5.13)	0.00	1.22 (0.65–2.28)	0.54	5.12 (2.63–9.96)	0.00	2.07 (0.94–4.57)	0.07
Grade								
1	1.00		1.00		1.00		1.00	
2	1.99 (0.90–4.44)	0.09	1.55 (0.67–3.58)	0.30	3.16 (0.99–10.09)	0.05	2.52 (0.77–8.25)	0.13
3	6.56 (3.22–13.37)	0.00	3.42 (1.49–7.83)	0.00	11.51 (3.96–33.41)	0.00	4.40 (1.34–14.46)	0.02
Pelvic LN metastasis								
Negative	1.00		1.00		1.00		1.00	
Positive	5.32 (3.07–9.22)	0.00	1.68 (0.65–4.35)	0.29	6.40 (3.27–12.52)	0.00	2.26 (0.63–8.17)	0.21
Paraaortic LN metastasis								
Negative	1.00		1.00		1.00		1.00	
Positive	2.99 (1.42–6.33)	0.00	1.01 (0.42–2.46)	0.98	2.64 (1.03–6.80)	0.04	0.51 (0.18–1.45)	0.20

DFS, disease-free survival; OS, overall survival; CI, confidence interval; HR, hazard ratio; RDW, red blood cell distribution width; LN, lymph node.

## Data Availability

The data pertaining to this study are available in the article.

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
