# Peer review of "Clinical Relevance of Red Blood Cell Distribution Width (RDW) in Endometrial Cancer: A Retrospective Single-Center Experience from Korea"

_cancers, 2023, doi:10.3390/cancers15153984_

Round 1

Reviewer 1 Report

The authors have presented results evaluating the relationship between preoperative red blood cell width (RBW) and survival outcome in 431 patients diagnosed with endometrial cancer between 2006-2018. RBW appears to be indicative of systemic inflammation and impaired immune dysregulation, lending biological plausibly to this relationship. This was a retrospective cohort study using electronic medical records from Yongin Severance Hospital. Uniform and rigorous inclusion criteria were applied to all patients. RBW was analyzed using the median value and women were classified as either having high or low RBW. The authors conclude that the results support patients with a high RBW had worse disease-free survival and worse overall survival. Although RBW has been shown to be related to outcomes with other cancers, this relationship has not been reported among endometrial cancer patients.

Specific comments are as follows:

1.      The statistical analysis section lacks a description concerning the analyses in Table 2. How were the hazard ratios in the crude of multivariable models calculated?

2.      The multivariable models did not include BMI, as possible confounder. Was this an oversite?

3.      Possible selection bias is mentioned in the discussion. It would be helpful if the authors would further discuss the possible implications related to their results.

Author Response

The authors have presented results evaluating the relationship between preoperative red blood cell width (RBW) and survival outcome in 431 patients diagnosed with endometrial cancer between 2006-2018. RBW appears to be indicative of systemic inflammation and impaired immune dysregulation, lending biological plausibly to this relationship. This was a retrospective cohort study using electronic medical records from Yongin Severance Hospital. Uniform and rigorous inclusion criteria were applied to all patients. RBW was analyzed using the median value and women were classified as either having high or low RBW. The authors conclude that the results support patients with a high RBW had worse disease-free survival and worse overall survival. Although RBW has been shown to be related to outcomes with other cancers, this relationship has not been reported among endometrial cancer patients.
Specific comments are as follows:
1.    The statistical analysis section lacks a description concerning the analyses in Table 2. How were the hazard ratios in the crude of multivariable models calculated?
→ Thank you very much for your valuable comments. 
 Concerning the statistical analysis section and the calculation of hazard ratios in Table 2, we apologize for the oversight in not providing a clear description. The hazard ratios presented in Table 2 were calculated using the Cox proportional hazards regression model. This information has now been included in the statistical analysis section to offer readers a better understanding of the analytical approach.

2.    The multivariable models did not include BMI, as possible confounder. Was this an oversite?

As you pointed out, BMI is a possible confounder in our multivariable models. We appreciate your keen observation, and you are correct that BMI is indeed an important covariate to consider in cancer survival studies. Various literature has demonstrated that cancer progression and survival outcomes are influenced by many factors such as myometrial invasion, CA-125, lympho-vascular space invasion, BMI, and preoperative leukocytosis), However, we did not look into these prognostic clinicopathological markers to focus on the clinical significance of RDW in the study.

3.      Possible selection bias is mentioned in the discussion. It would be helpful if the authors would further discuss the possible implications related to their results.

We acknowledge the reviewers' comment regarding the mention of possible selection bias in our discussion. We have now expanded the discussion section to elaborate on the implications of this potential bias on our results. While we strived to apply uniform and rigorous inclusion criteria, we recognize that retrospective cohort studies may be susceptible to certain biases. We have emphasized the need for cautious interpretation of our findings, especially in light of this limitation, and have highlighted the importance of further research to validate and corroborate our results.

Reviewer 2 Report

This paper by Kyung-Jin et al. is an interesting and simple approach to provide another prognosis factor. Concerning higher stages of endometrial cancer the results could be of value. It is no factor to contribute to treatment decisions.

It would be interesting to correlate these findings to for example TNF alpha and IL-6 with regard to inflammatory backgrounds. Moreover, it should be of utmost importance to exclude definitely other disorders impairing hematopoesis.

Author Response

This paper by Kyung-Jin et al. is an interesting and simple approach to provide another prognosis factor. Concerning higher stages of endometrial cancer, the results could be of value. It is no factor to contribute to treatment decisions.

It would be interesting to correlate these findings to for example TNF alpha and IL-6 with regard to inflammatory backgrounds. Moreover, it should be of utmost importance to exclude definitely other disorders impairing hematopoesis.

→ Thank you very much for your valuable comments.

Reviewer 3 Report

With pleasure, I read the paper titled: “Clinical relevance of red blood cell distribution width (RDW) in endometrial cancer”. Overall, the paper reads very well. The novelty and major strength lies in being the first study to examine the role of RDW in endometrial cancer. Collectively, the article is well-written with good flow of ideas, proper English language, up-to-date citations, and focused tables and figures. The authors adequately introduced the topic and provided rationale for the study. The authors are commended for briefly enlightening the readers about molecular mechanisms for the crosstalk between RDW changes and cancer. The methods section is detailed enough to allow for reproducibility and transparency. The results are written in simple way and data are summarized in pertinent tables and figures. The discussion section summarized the principal findings, provided discussion of the key results, and entertained comparison with published literature. The conclusion is in line with the reported results. I have the following comments:

1. Title. I recommend adding the following at the end for completion “a retrospective single-center experience from Korea” to highlight the type of study and its origin.

2. Abstract. The total number of sample size should be reported (n=431) and similarly the numbers of patients belonging to low and high RDW groups. The conclusion should be revised to reflect the impact of RDW on recurrence as well as survival. The last sentence should be omitted or revised to something like “prospective studies are needed to validate the findings of the current study”. In the Simple Summary, the authors used the term progression-free survival (PFS) interchangeably with disease-free survival (DFS), which is not true. The authors need to use only DFS since this what they have evaluated in their study.

3. Introduction. Please clearly indicate the significance of the study by highlighting the present study is the first-ever investigation to examine the impact of RDW among endometrial cancer patients.

4. Methods. I have a small concern regarding the definition of high vs low RDW groups. The authors defined the groups based on the median cut-off value, which can be skewed according to the selected population. However, I wonder if it would have been much accurate if the cut-off was determined based on the normal range. Meaning, on average, the normal range of RDW is 12-16% in adult females, and therefore high RDW groups might have defined based RDW greater than 16%. The authors need to comment on this aspect. Additionally, for other studies, did the authors of the other studies in other cancers defined RDW groups based on the median values?

5. Results. Have the authors looked into other important prognostic clinico-pathological markers (such as myometrial invasion, CA-125, lympho-vascular space invasion, and preoperative leukocytosis). Despite the RDW was significant independent factor of 5-year OS (89% vs 92% in high and low RDW groups, respectively) STATISTICALLY, however, the difference does not seem significant CLINICALLY. The authors may need to highlight this observation. For figure 2, I encourage to use color coding for better visualization as well as adding the value percentages directly into the figures for better accessibility to data.

6. Discussion. RDW is a hematological parameter. Please include a small paragraph highlighting the prognostic significance of other hematological parameters in endometrial cancer (such as preoperative leukocytosis, anemia, and thrombosis) using high-quality research designs such as systematic review and meta-analysis reports of published literature. Examples include: PMID: 34298450, PMID: 33529973, etc. The authors need to provide a small paragraph about the future directions, for example, the need to compare sensitivity/specificity of RDW as a prognostic marker. Lastly, the authors need to acknowledge additional limitations such as the single-center experience (hence impacting the generalizability of the findings), the single-group design limits the authors’ ability to determine cause-and-effect correlations, and not all patients had 5-year follow-up.

7. Conclusion. The authors should include a sentence that says, “additional studies are needed to validate the findings of the current one”.

Round 2

Reviewer 2 Report

Thank you for the revision. The paper is ready for publication.